# Radar vision in the mapping of forest biodiversity from space

Soyeon Bae [1]*, Shaun R. Levick [2,3], Lea Heidrich[1], Paul Magdon [4], Benjamin F. Leutner [5], Stephan Wöllauer [6], Alla Serebryanyk[7], Thomas Nauss [6], Peter Krzystek[7], Martin M. Gossner [8], Peter Schall[9], Christoph Heibl [10], Claus Bässler[10,11], Inken Doerfler[11,12], Ernst-Detlef Schulze[13], Franz-Sebastian Krah [14,10], Heike Culmsee [15], Kirsten Jung [16], Marco Heurich[10,17], Markus Fischer[18,19], Sebastian Seibold [11,1], Simon Thorn [1], Tobias Gerlach[20], Torsten Hothorn[21], Wolfgang W. Weisser [11] & Jörg Müller [1,10]

Recent progress in remote sensing provides much-needed, large-scale spatio-temporal information on habitat structures important for biodiversity conservation. Here we examine the potential of a newly launched satellite-borne radar system (Sentinel-1) to map the biodiversity of twelve taxa across five temperate forest regions in central Europe. We show that the sensitivity of radar to habitat structure is similar to that of airborne laser scanning (ALS), the current gold standard in the measurement of forest structure. Our models of different facets of biodiversity reveal that radar performs as well as ALS; median $R^2$ over twelve taxa by ALS and radar are 0.51 and 0.57 respectively for the first non-metric multidimensional scaling axes representing assemblage composition. We further demonstrate the promising predictive ability of radar-derived data with external validation based on the species composition of birds and saproxylic beetles. Establishing new area-wide biodiversity monitoring by remote sensing will require the coupling of radar data to stratified and standardized collected local species data.

[1] Department of Animal Ecology and Tropical Biology, University of Würzburg, 97074 Würzburg, Germany. [2] CSIRO Land and Water, PMB 44, Winnellie 0822 NT, Australia. [3] College of Engineering, IT, and Environment, Charles Darwin University, Winnellie 0909 NT, Australia. [4] Forest Inventory and Remote Sensing, Faculty of Forest Sciences and Forest Ecology, University of Göttingen, Büsgenweg 5, 37077 Göttingen, Germany. [5] German Remote Sensing Data Center (DFD), Earth Observation Center (EOC), German Aerospace Center (DLR), 82234 Weßling, Germany. [6] Faculty of Geography, Philipps-University Marburg, Deutschhausstr. 10, 35037 Marburg, Germany. [7] Department of Geoinformatics, Munich University of Applied Sciences, 80335 München, Germany. [8] Forest Entomology, Swiss Federal Research Institute WSL, Zürcherstrasse 111, CH-8903 Birmensdorf, Switzerland. [9] Silviculture and Forest Ecology of the Temperate Zones, University of Göttingen, Büsgenweg 1, 37077 Göttingen, Germany. [10] Bavarian Forest National Park, Freyunger Str. 2, 94481 Grafenau, Germany. [11] Terrestrial Ecology Research Group, Department of Ecology and Ecosystem Management, Technical University of Munich, Hans-Carl-von-Carlowitz-Platz 2, 85354 Freising, Germany. [12] Institute of Biology and Environmental science, Vegetation science & Nature conservation, University of Oldenburg, Ammerländer Heerstraße 114-118, 26129 Oldenburg, Germany. [13] Max Planck Institute for Biogeochemistry, Box 10016407701 Jena, Germany. [14] Plant Biodiversity Research Group, Department of Ecology & Ecosystem Management, Technical University of Munich, Emil-Ramann Strasse 2, 85354 Freising, Germany. [15] DBU Natural Heritage, German Federal Foundation for the Environment, An der Bornau 2, 49090 Osnabrück, Germany. [16] Evolutionary Ecology and Conservation Genomics, University Ulm, Albert-Einstein-Allee 11, 89069 Ulm, Germany. [17] Chair of Wildlife Ecology and Wildlife Management, University of Freiburg, 79085 Freiburg im Breisgau, Germany. [18] Institute of Plant Sciences, University of Bern, 3012 Bern, Switzerland. [19] Senckenberg Biodiversity and Climate Research Centre (SBiK-F), 60325 Frankfurt am Main, Germany. [20] UNESCO-Biosphere Reserve Rhön, Oberwaldbehrunger Str. 4, 97656 Oberelsbach, Germany. [21] Epidemiology, Biostatistics and Prevention Institute, University of Zurich, Hirschengraben 84, 8001 Zürich, Switzerland. *email: soyeon.grace.bae@gmail.com

The impact of humans on the planet has progressively escalated to the extent that the current geological age is referred to as the Anthropocene, in recognition of the geological dimensions of the human footprint[1]. Its characteristics include declines in non-human populations and the extinction of species, due most prominently to anthropogenically mediated habitat degradation[2,3]. The resulting loss of biodiversity is evident at local and landscape scales, but attempts to measure habitat loss for diverse species over large areas have been frustrated by the high cost and the considerable effort involved. The unsolved challenge here is to monitor area-wide diversity within and between habitats (α-diversity and β-diversity, respectively) across large areas, which impedes estimates of total diversity (γ-diversity) of landscapes. However, over the last decade, advances in remote sensing have led to an exponential increase in the use of these technologies, including in ecological investigations[4,5], and a recognition of their potential in obtaining reliable and frequent updates on the spatial information required to monitor biodiversity over larger areas, information that is essential for conservationists. Skidmore et al.[6] called upon ecologists and space agencies throughout the world to forge a global-monitoring strategy that includes a definitive set of biodiversity variables and a plan for tracking them from space. Despite the theoretical progress that has been made under the umbrella of Group on Earth Observation Biodiversity Observation Network (GEO-BON), quantitative evidence of how well different essential biodiversity variables[7] can be mapped and monitored from space is lacking, and the relationship of these variables to different facets of biodiversity remains poorly understood[8].

With its ability to characterise the complex three-dimensional (3-D) structure of terrain and vegetation, airborne laser scanning (ALS) has been particularly successful in biodiversity monitoring[9]. Objective remote measurements can now be conducted with ALS and the acquired data used to model vegetation metrics (e.g. canopy cover, height, layering, and basal area) that traditionally were estimated based on laborious fieldwork. The 3-D data acquired with ALS has provided the basis for a number of advances in animal ecology and biodiversity conservation[10,11]. Although large-area mapping by space-borne laser scanning has thus far been limited in scope, progress towards this long-term goal is being made by programmes such as Global Ecosystem Dynamics Investigator (GEDI)[12] and ICESat-2 (Ice, Cloud, and land Elevation Satellite-2)[13], in which spot measurements of canopy height and profile layering are obtained within the laser beam footprint (~22 and 90 m, respectively). Both missions are expected to supply critical information in support of the mapping of structural essential biodiversity variables. While current and future space-borne laser-scanning systems provide only patchy information, space-borne synthetic aperture radar (SAR) systems are also sensitive to the geometric properties of the Earth's surface, such as forest canopy structure, and capable of complete coverage of the entire globe. Hence, SAR data could be an alternative source for ecologically meaningful information on vegetation structure from regional to global scales. SAR is similar to ALS as both remote sensing techniques actively emit electromagnetic radiation and measure the returned signal. A major advantage of SAR is its ability to penetrate clouds, making it a suitable technique also for regions with nearly constant cloud coverage, such as the tropics or mountain areas. Depending on the wavelength used (e.g. C-band), SAR backscatter signals can be interpreted to derive ecologically meaningful structural information from terrain and vegetation. SAR has already had a significant impact on ecological research, and both C-band and L-band sensors have been used extensively in the mapping of biomass within boreal[14] and tropical forest regions[15,16].

The launch of the Sentinel-1 mission, a constellation of two C-band SAR satellites, by the European Space Agency in 2014 and 2016 revolutionised SAR remote sensing, due to Sentinel 1's unprecedented combination of high spatial resolution (5–20 m), high revisit frequencies (5–10 days), complete geographic coverage and the ESA's open-access policy regarding the availability of the collected data. C-band SAR has a wavelength of 5.6 cm, which means it is sensitive to vegetation structures and is likely to be scattered from elements within the tree canopy. However, the formation of SAR backscatter signals is complex, as factors other than canopy structure, such as scan angle, direction, soil moisture and plant water content, also exert considerable influence on backscatter properties. Some of these confounding influences can be mediated by making use of multi-date acquisitions, as was the case in this study.

Using the open-access, dense time-series data obtained by the Sentinel-1 mission, we conduct the first evaluation of Sentinel-1's potential in biodiversity mapping. Our study begins with a comparison of the ecological application of radar (henceforth, "Sentinel-1" is referred to as "radar") metrics vs. the well-established ALS metrics in providing a better understanding of habitat structure in forest ecosystems. A suite of ground-truth taxonomic and phylogenetic biodiversity measures covering within forest stand (α-) and among forest stand (β-) diversity from a broad range of trophic levels and taxa (henceforth "functional groups") is then modelled using either ALS data or time-series radar data to explore the extent to which rich time-series radar data can be used to represent ecologically meaningful gradients of habitat conditions in temperate forests. Thus, we quantify the predictive power of radar in modelling different aspects of biodiversity, including species composition and richness and phylogenetic diversity, and compare the results to those obtained using very high density (8–40 pulses/m² in this study) ALS data. For this purpose, we make use of a distributed ground-based network of 463 biodiversity monitoring plots spanning five Central European temperate forest regions and capturing biodiversity data for 12 functional groups. Finally, to test their suitability for biodiversity mapping and monitoring, the radar models for two taxa are validated using independent external data collected from areas outside the five training areas.

Our analysis shows the close association of the structural attributes of forests as described by radar and ALS data, which also similarly reflect gradients of forest maturity and structural heterogeneity. As predictors of biodiversity, the two remote-sensing techniques are similar in their power, albeit with radar data being superior for species composition and ALS for species richness. Global biodiversity monitoring requires both a consistent method of satellite image acquisition and open access to those images. Our study demonstrates the potential of such data for monitoring biodiversity of forests and thus of other large-scale habitats as well.

## Results

**The ecological relevance of radar-derived variables.** Canonical correlation analysis (CCoA) showed a strong correlation between the habitat metrics derived from the ALS and radar sensors. The ecological relevance of the latter with respect to 3-D forest structure and resident animal diversity was established in prior studies[10]. Among the 13 canonical axes from the two remote-sensing data sets, nine statistically significant pairs ($p < 0.05$ with Pearson's correlation test), explaining 96.30% of the variance of the datasets (Supplementary Table 4), were identified. The first and second axes showed the highest canonical correlation coefficients (for the pairs of canonical axes from the two datasets): 0.92 and 0.75, respectively (Fig. 1).

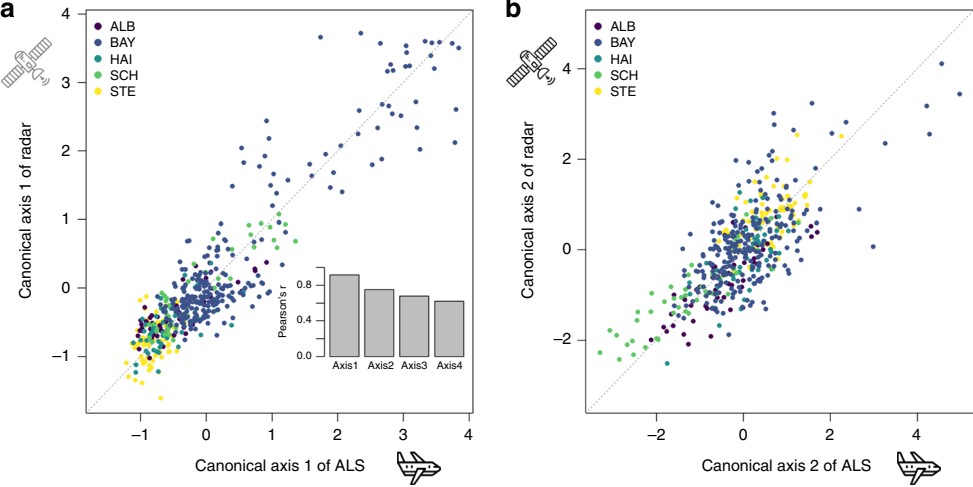

**Fig. 1** Correlations between the metrics from the two sensors. Correlations between the **a** first (forest maturity) and **b** second (structural heterogeneity) axes of a canonical correlation analysis, extracted to maximise the correlation between the data sets of the radar- and airborne laser scanning (ALS)-derived variables. The inset in (**a**) shows the Pearson's correlation coefficients for the first four axes of the two data sets with the significance level $p < 0.001$. Source data are provided as a Source Data file

The first canonical axes represented a gradient of decreasing forest maturity, as the first ALS axis correlated negatively with both the penetration ratio of the canopy layer and the vegetation height and positively with the gap area described by the ALS metrics (Fig. 2 and Supplementary Table 5). The correlation of the first radar axis with yearly and winter radar backscatter was highly negative (Fig. 2 and Supplementary Table 6). The second canonical axes represented the variability in height, as the second ALS axis had a strong correlation with the standard deviations of vegetation height and canopy surface height. Similarly, the second radar axis correlated with the standard deviation of the yearly and summer backscatter, which we calculated to represent structural heterogeneity indices. The third and fourth canonical axes represented gradients of structural heterogeneity, as the third ALS axis strongly correlated with the penetration of the regeneration layer and the edge length of forest gaps and the fourth ALS axis with foliage height diversity. The third radar axis showed strong correlations with texture measures (contrast and orderliness, quantifying spatial heterogeneity) and the fourth radar axis with the ratio between the two descriptors of polarisation, VV (vertically transmitted, vertically received radar pulses) and VH (vertically transmitted, horizontally received radar pulses).

**The drivers of different components of diversity.** Boosted generalised additive models (GAMs), i.e. fixed effects models, were employed with five-fold cross-validation for the internal validation of all response variables: the main axes of species assemblage composition based on non-metric multidimensional scaling (NMDS) ordination scores, log-transformed species richness and phylogenetic diversity, with the latter calculated as the standardised effect size to ensure independence from species richness. Overall, the performances of the radar and ALS metrics were similar. In addition, for both sensors, with the use of metrics related to forest maturity the assemblage composition was better predicted than were diversity indices. This was indicated by the cross-validated $R^2$ (coefficient of determination) and root mean square error (RMSE) values for the first NMDS axes representing assemblage composition (median $R^2$ values over 12 functional groups by ALS and radar: 0.51 and 0.57, respectively) and the second NMDS axes representing assemblage composition (0.30

and 0.27), species richness (0.21 and 0.11) and phylogenetic diversity (0.19 and 0.16) (Fig. 3 and Supplementary Table 7; additional RMSE results are shown in Supplementary Table 8). The first axes of assemblage composition (NMDS1) were distinctively better predicted by radar than by ALS, with the exception of the assemblage composition of bats, but the second axes of assemblage composition (NMDS2) were better predicted by ALS, with the exceptions of the assemblage composition of lichens and phytophagous beetles. However, for species richness and phylogenetic diversity, the predictive performances of the two sensors were comparable. To check the robustness of our results of arthropods for sample size we reanalysed the data on a subset of plots with sufficient sample completeness. These findings corroborated the findings of the total data set (Supplementary Fig. 22).

To take into account repeated measures within the five forest regions, we additionally fitted mixed effects models in which region was a random factor (see the "Methods" section for details). Overall, this reduced the explained variance in the ALS and radar models, but the results between taxa were highly variable. The decrease in the explained variance was strongest in ground-living spiders and carabids, although a decrease in that of bats was obtained as well. The loss of explained variance by a region effect in the mixed effects models was stronger in the ALS models (0.14, a median of 48 response variables) than in the radar models (0.10) (Supplementary Tables 9 and 10). This tendency of the superiority of one sensor over the other in the mixed effects models was mostly consistent with the tendency in the fixed effects models, except in the cases of 4 of 48 response variables.

The most important variables in the predictions of the NMDS axes representing assemblage composition were the penetration ratio of the canopy-understorey ($PR_{h>2m}$) in the ALS model and the winter VH ($VH_{winter}$) in the radar model. This was determined consistently across 12 functional groups (Supplementary Figs. 6–17). In the ALS models, $PR_{h>2m}$ was the most important predictor of the first and second NMDS axes of 11 of the 12 functional groups, with the exception being the assemblage composition of necrophagous beetles. Using the same approach for the radar models, $VH_{winter}$ was the most important predictor of 10 of the 12 groups, with the exceptions being the assemblage compositions of carabid beetles and bats. Nevertheless, among the exceptional groups, $PR_{h>2m}$ and $VH_{winter}$ ranked second among

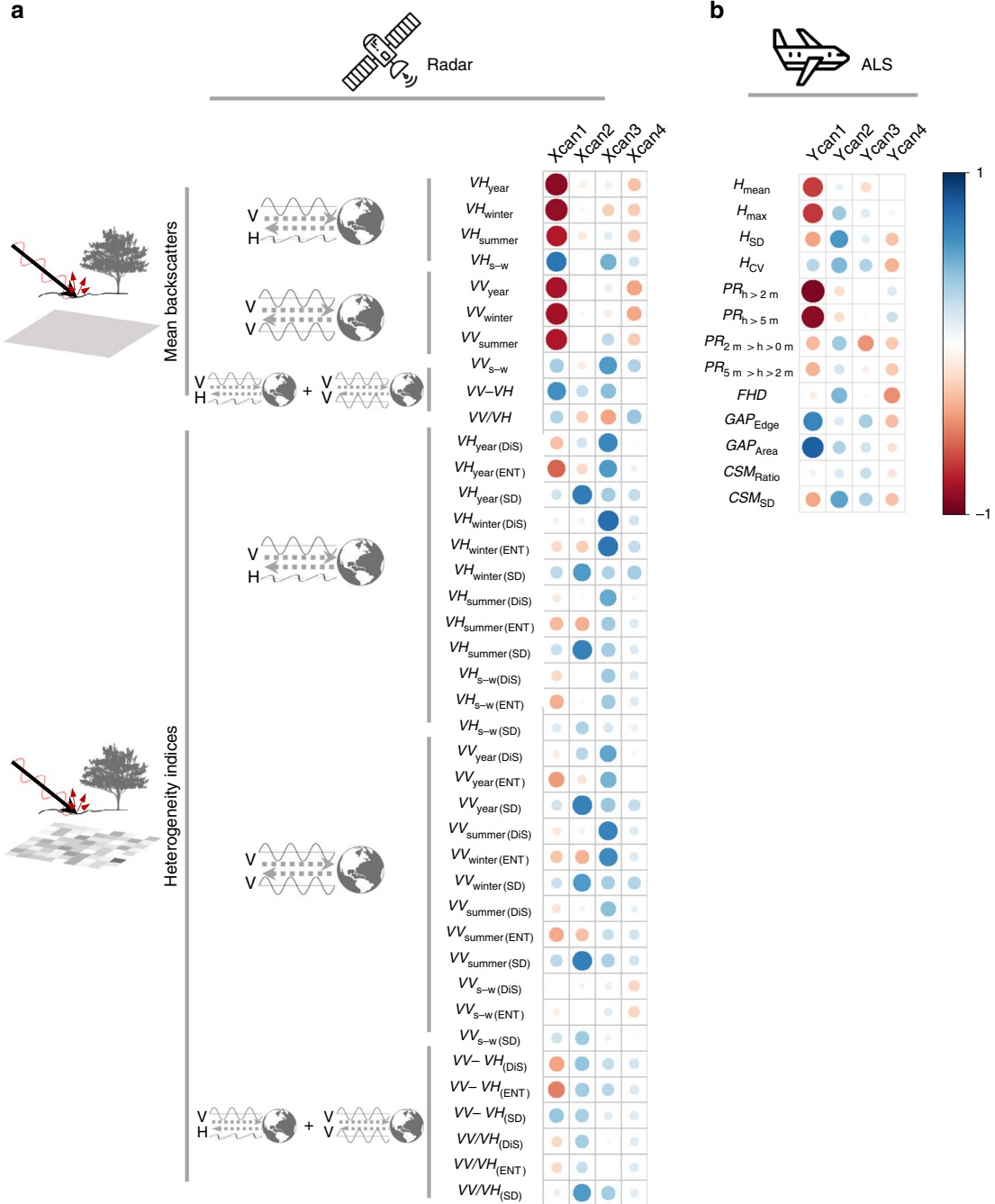

**Fig. 2** Ecological relevance of the metrics from the two sensors. Pearson's correlation matrix **a** between the first four main canonical axes (Xcan1–4) and the radar metrics and (**b** between the first four main canonical axes (Ycan1–4) and the ALS metrics at the significance level $p < 0.05$. The first canonical axes represent a gradient of decreasing forest maturity, and the second, third, and fourth canonical axes gradients of structural heterogeneity. Positive correlations are displayed in blue and negative correlations in red colour. Colour intensity and the size of the circle are proportional to the correlation coefficients. See Supplementary Tables 4 and 5 for details

the list of dominant factors in the ALS models of necrophagous beetles and in the radar models of bats, respectively.

The most critical predictors of species richness varied across the different functional groups. The predictors related to structural heterogeneity as identified by ALS, such as the coefficient of variation of vegetation height ($H_{CV}$), the standard deviation of canopy surface height ($CSM_{SD}$) and the edge length of forest gaps ($Gap_{Edge}$), actively contributed to the construction of the species richness models. Likewise, in the species richness models derived from radar data, horizontal heterogeneity predictors, such as standard deviation, dissimilarity and the

entropy of radar backscatters, was of greater importance than in the corresponding assemblage composition models. In contrast to the species richness models, the phylogenetic diversity models derived from ALS and radar were strongly driven by measures sensitive to forest maturity, such as $PR_{h>2m}$ and $PR_{2m>h>0m}$ and $VH_{winter}$, respectively.

External validation of the assemblage composition models of two selected groups, birds ($n = 72$) and saproxylic beetles ($n = 91$), using data from outside the five forest regions further demonstrated the substantial predictive power of the models (coefficients of determination: 0.26 and 0.22, respectively) (Fig. 4).

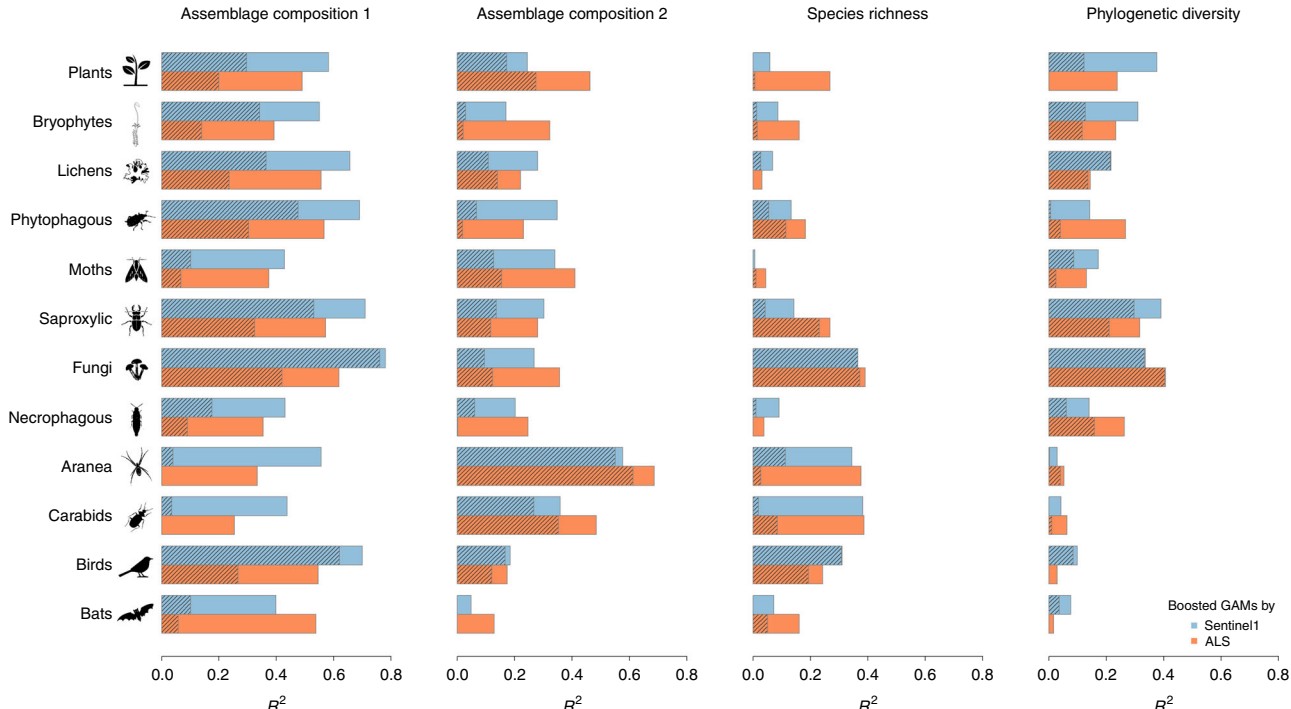

**Fig. 3** Predictive power of radar and ALS in modelling different aspects of biodiversity. Cross-validated performance ($R^2$, coefficient of determination) of assemblage habitat models (boosted generalised additive models), i.e. fixed effects models, using the ALS (orange bars) and radar (blue bars) data sets. The shaded bars represent $R^2$ derived from the mixed effects models. For mixed effects models, $R^2$ was calculated using only the fixed factors to predict the response variables, in order to exclude the variance explained by region

However, some of the validation plots featured very different bird species composition than those from the training space, such as those of the Rhön region (Fig. 4a and Supplementary Figs. 18a and 19a), and, as is to be expected, could not be predicted well in terms of NMDS1.

## Discussion

Our results showed a strong correlation between pairs of canonical axes from radar and ALS data describing gradients of forest maturity and structural heterogeneity in forest ecosystems on a 1-ha scale. In biodiversity models of 12 functional groups, radar and ALS performed equally well. While the model performance of radar was better than that of ALS in predicting species composition, ALS was better in predicting species richness and phylogenetic diversity. The results obtained with both sensors showed the closer association of species composition and phylogenetic diversity with gradients of forest maturity, and that of species richness with structural heterogeneity. However, the models of the diversity indices were inferior to those of assemblage composition.

The prediction accuracy of ALS for the structural attributes of forests and consequently for attributes of the associated communities, such as taxonomic diversity, has been well established over the last two decades[10]. Previous studies comparing ALS and radar in terms of the accuracy of forest attribute estimation for variables, such as canopy height, stem volume and biomass revealed the superiority of ALS over radar at the local scale[17,18]. ALS was also shown to be better for high-accuracy characterisations of understorey layering and the structural complexity at local scales[19,20]. These findings are not surprising, given the small footprint and the available high-energy sources of airborne platforms, compared with the challenges of interpreting the longer wavelengths and larger footprints of space-borne C-band

SAR[21]. Nonetheless, while at the scale of individual trees radar may not be able to provide the same level of height accuracy provided by ALS, this does not preclude the possibility that backscatter properties recorded from space can suitably capture the broader structural properties relevant to forest-dwelling organisms.

Using a similar CCoA methodology, field inventory data were previously compared with ALS data to determine the ability of the latter to predict critical forest stand structure and animal communities[11,22]. In their CCoA analysis of ALS and forest field inventory data, Lefsky et al.[22] found that forest structure could be characterised by three main factors: aboveground biomass (related to height), canopy cover (or openness) and structural heterogeneity (related to height variability). Our CCoA analysis of radar and ALS metrics showed that two main factors, forest maturity and structural heterogeneity, comprehensively captured forest structure. The first pair of axes was directly related to forest maturity, which was represented by canopy cover (or openness) and canopy height. The fusion of these axes was likely due to the lower penetration depth of C-band SAR, which is limited to the very upper layers of the canopy, than of L-band and P-band systems using longer wavelengths. The relatively short wavelengths of C-band SAR account for its poor ability in separating canopy height or biomass from canopy cover. A weak correlation between aboveground biomass and the backscatter of C-band SAR in dense forest has been reported and is generally associated with the early saturation of backscatter intensities for high aboveground biomass[23,24]. The remaining significant axes from CCoA mostly reflected the structural heterogeneity of the forest stand, which is in line with the third main characteristic of forest structure identified by Lefsky et al.[22]. The ALS and radar metrics approximating vertical and horizontal heterogeneity were associated with those structural heterogeneity axes. Image texture was employed in previous studies to improve land cover

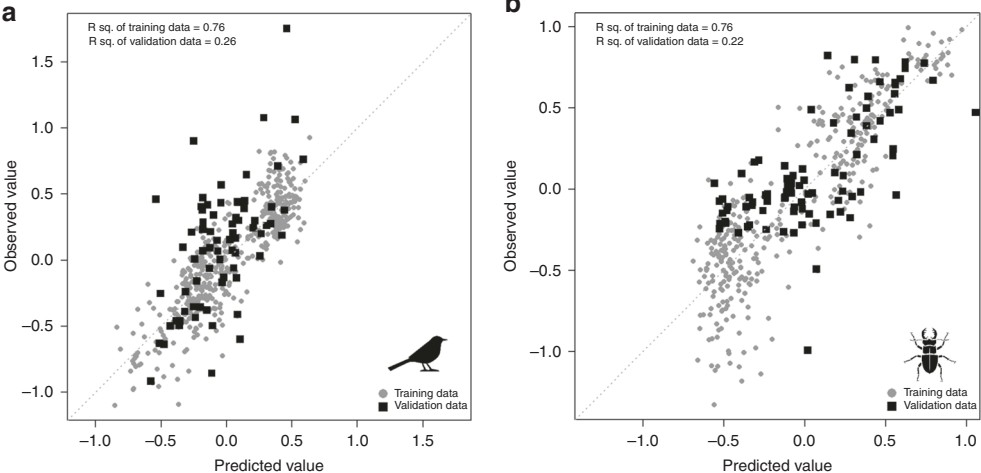

**Fig. 4** External validation of the assemblage composition models. Scatter plots of the first axis of the NMDS of **a** birds and **b** saproxylic beetles, between the observed and predicted value of the training data (grey circles) and the validation data (black squares). The $R^2$ (the coefficient of determination) of training data and validation data are shown. Source data are provided as a Source Data file

classification[25] but it had not been used to characterise the structural heterogeneity of a forest. Our study demonstrated that, by using image texture, the structural heterogeneity of a forest at a 1-ha spatial level and a resolution of 10 m can be captured by C-band radar, despite its limited penetration of the canopy.

The CCoA also demonstrated that, for biodiversity modelling, multi-temporal radar data can substitute for a large proportion of the information available from ALS. Moreover, compared to readily interpretable ALS metrics, we were able to derive an ecological interpretation from numerous radar metrics, which was a prerequisite for the later biodiversity modelling.

Both the changes in openness as a consequence of forest succession and the accompanying changes in microclimatic conditions heavily structure species composition[26–30]. These accumulated effects associated with forest maturity were well reflected in the radar as well as the ALS models, as both showed the strong effects of forest maturity on the species composition of most of the 12 functional groups, even after controlling for repeated measurements in a specific region. Interestingly, the model performance of radar in predicting species composition was better than that of ALS. Although radar cannot provide the highly detailed information on forest structure that is generated by ALS, its measurements still allow a sufficiently fine-scale description of forest maturity. Moreover, multi-temporal radar data had a better discriminatory ability with respect to the composition of dominant tree species than did ALS data on single leaf-on acquisition (see Supplementary Fig. 21 for additional models predicting the conifer tree ratio). The split between conifers and broadleaved trees greatly affects the composition not only of herbivores but also of fungi[31] and may cascade to higher trophic levels[32,33]. Similarly Schaffers et al.[34] found that plant composition is a more powerful predictor of the communities of predators and herbivores than is the physical habitat structure of grasslands. Therefore, radar data combine two important determinants of forest assemblage composition: maturity and conifer proportion. The drop in predictive power regarding the assemblage composition of carabids, spiders and bats after accounting for regions may reflect the geographical patterns or unmeasured environmental determinants of these groups. In the case of bats, the number of species found in Germany is highly regionally restricted (see Supplementary Fig. 5). For ground-dwelling carabids and spiders, climate and soil conditions, mostly related to regional differences, likely override the structural patterns derived from remote-sensing data. However, within a specific region, remote-sensing data well predicted both ground-dwelling beetles and spiders[11,35].

Overall, the models of species richness and phylogenetic diversity were inferior to those of assemblage composition. This was not surprising, as species richness does not consider the taxonomic identity of species and ignores species turnover. For example, two forest patches with different environmental conditions, such as open vs. closed forests, might exhibit the same species richness but harbour assemblages that differ completely in their composition. This was previously shown for saproxylic beetles, in which large numbers of conifer specialists were present in open forests whereas broadleaf specialists predominated in forests characterised by a closed canopy[36]. Hence, in predictions of species richness and assemblage composition based on environmental information, that is, on remote-sensing data, the relationship between predictor and response is much weaker for the former than for the latter. This was also suggested by Leutner et al.[28], who found that plant assemblage composition, but not species richness, could be successfully modelled with ALS and hyperspectral data. Along the same lines, a recent study reported an overall weak and highly variable relationship between species richness and carbon stock at the stand scale in the temperate forests of Europe, presumably because of trade-offs between species[37].

However, we showed that various structural heterogeneity indices in both the ALS and radar metrics strengthened the respective species richness models. The structural heterogeneity metrics of ALS were important for the species richness of most groups (Supplementary Figs. 6–17). This is consistent with the habitat-heterogeneity hypothesis, which assumes increasing species richness with the increase in niche availability arising from habitat heterogeneity[38]. Support for this assumption comes from previous ALS studies in which the strong effects of 3-D structural heterogeneity on species richness were described[10]. In our approach, radar-derived data were used to derive heterogeneity metrics in forests that were then applied successfully in biodiversity modelling. These metrics appeared to have been those representing the most important drivers in the species richness models of several taxonomic groups. Hence, our study well demonstrates the high explanatory and predictive power of

coarse, space-borne, radar-derived data in ecological studies at local scales.

Forest attributes not only differentiate between communities, but they also can be used to recognise the phylogenetic diversity within communities. For example, forest succession has been shown to shift communities in terms of their phylogeny, from closely related species in early-stage forests to more remotely related species in mature forest stands[39]. This finding was supported by our study, which showed the dominance of forest maturity in the phylogenetic models whose performance for several groups was moderate, such as bryophytes, saproxylic beetles and fungi. However, the phylogenetic diversity of higher trophic groups was not well predicted, as the determinants of those models were associated with the density of the understorey (Supplementary Figs. 6–17). Species richness and phylogenetic diversity patterns in forests may follow very different and at times even opposite patterns[40,41]. Hence phylogenetic diversity is not a surrogate but a complementary biodiversity measure that provides additional information on local diversity. However, the high $R^2$ of the coefficient of determination determined using the radar-based models for both the phylogenetic diversity of some groups, such as plants and saproxylic beetles, and the species composition support the global monitoring of phylogenetic diversity from space, and not only for plants[42].

The results of our studies of five forest regions in Central Europe demonstrate the potential of open-access space-borne radar data in predicting different components of biodiversity. Importantly, the correlation between radar and ALS gradients indicated the substantial cost-effectiveness of Sentinel-1 approach when applied to large-area mapping. As evidenced by the attributes of Sentinel-1 backscatters in their representation of forest maturity and tree composition, two of the main drivers of local species turnover, and by the various measures of structural heterogeneity, open-access Sentinel-1 clearly offers an alternative method to model the biodiversity of different functional groups. Furthermore, as gamma diversity could be estimated as a product of alpha and beta diversity, Sentinel-1 can be applied to estimate gamma diversity even for large landscapes where ground estimations will stay impossible.

The shortcoming of Sentinel-1 data that we uncovered was in the prediction of species richness and phylogenetic diversity for groups that were more strongly driven by the development of the understorey. Weak permeability through overstorey layers is unavoidable with space-borne C-band SAR systems, due to their short wavelengths[21], whereas L-band and P-band systems make use of longer wavelengths. Nonetheless, Sentinel-1 performed as well as ALS with respect to the biodiversity models of groups driven by forest maturity or specific indices of structural heterogeneity. In the near future, L-band and S-band SAR data will become increasingly availability (e.g. NASA-ISRO synthetic aperture radar (NISAR), a dual-frequency SAR carried on an Earth observation satellite). Used in conjunction with Sentinel-1's C-band, they may allow a better characterisation of the understory and of the different-sized elements of forest structure.

Although our study covered five forested regions, these were representative only of the major temperate forest ecosystems of Central Europe. Nonetheless, given that radar data for the Earth's forested regions are ubiquitously available, there is more than ample opportunity to test the generality of our findings in essentially all forests. The major barrier to the larger-scale application of our methodology is the lack of availability of georeferenced and well-stratified (both spatially and ecologically) biodiversity data that span multiple taxa. Datasets such as those from the Biodiversity Exploratories[43] together with those generated in well-established long-term biodiversity monitoring, such as undertaken at the Steigerwald[44] and in the Bavarian Forest

National Park[45] at the scale of the individual forest, provide excellent test-dataset allowing the broader application of the approach described herein. However, at larger scale such as at the country level or within the EU as a whole, standardised monitoring systems with high resolution are currently available only for a few taxa, for example, bird surveys by the Umbrella Organization of German Avifaunists (Dachverband Deutscher Avifaunisten, DDA). Until governments compile or generate data from the well-stratified, consistent sampling of a larger number of taxonomic and trophic levels, the immediate application of Sentinel-1 data will be restricted to existing data, such as the DDA's bird data. For forests, Sentinel-1 data may well be highly suitable for the mapping of environmental gradients at national scales, which in turn can facilitate the stratified random selection of appropriate locations for field-based biodiversity calibration and validation sites, e.g. selected sites from national forest inventories. For generations, biodiversity data have been collected according to a bottom-up approach. However, the tools to complement this information, by analyses conducted on top-down-collected data, are now available. Their use will ensure that a broad spectrum of biodiversity is covered. Our research has shown a way forward in the mapping of structural indicators of biodiversity as determined from space. Yet, the question remains: how well can changes in biodiversity be monitored? Since radar provides multi-temporal measurements needed to detect trends, it has the potential to provide a basis for future research. Furthermore, thresholds for the detection of alterations in habitat conditions that trigger positive and negative biodiversity outcomes, the time delay in extinction after the habitat degradation and synergistic process with other threats such as climate change must still be defined.

## Methods

**Study site.** The study was conducted at up to 463 plots in five forest regions distributed from north to south in Germany and representative of forest habitat types in Central Europe (Supplementary Fig. 1). The data were compiled from three different projects: Biodiversity Exploratories[43], the BIOKLIM Project[45] and the Steigerwald Project[44,46]. Data were collected from 150 experimental plots of the Biodiversity Exploratories study site. These had been established in three regions: (1) 50 plots in the UNESCO Biosphere Reserve Schorfheide-Chorin, (2) 50 plots in the National Park Hainich and its neighbouring areas and (3) 50 plots in the UNESCO Biosphere Reserve, Schwäbische Alb. From the BIOKLIM Project, conducted in the Bavarian Forest National Park, 244 plots among the 293 sampling plots were selected; the 49 excluded plots were those in which the change in canopy cover between 2007 (year of ALS acquisition) and 2016 (year of radar acquisition) exceeded 20% due to disturbances such as bark beetle infestation and windthrow[47]. From the Steigerwald Project, located in the Steigerwald forests in Bavaria, 69 plots were included. For the analysis of each functional group, plots for which observations of the corresponding group were available were selected. The number of investigated plots per group was 454 for plants, 298 for bryophytes, 290 for lichens, 362 for phytophagous beetles, 219 for moths, 361 for saproxylic beetles, 458 for fungi, 361 for spiders, 347 for carabids, 334 for necrophagous beetles, 456 for birds and 201 for bats.

The Schorfheide-Chorin region (SCH) is located in the lowlands (80–140 m above sea level, a.s.l.)) of north-eastern Germany (N 52°86′–53°19′; E 13°63′–14° 00′). It is a glacially formed landscape with many wetlands. The Hainich region (HAI) is located in the hilly areas (330–550 m a.s.l.) of central Germany (N 51° 05′–51°37′; E 10°21′–10°53′). The Schwäbische Alb (ALB) region is located in the low mountain range (740–870 m a.s.l.) of south-western Germany (N 48°34′–48°50′; E 9°20′–9°50′). The three regions of the Biodiversity Exploratories cover different forest management intensities: unmanaged old-growth forests, managed uneven-aged forests and managed age-class forests including natural broad-leaved tree species, mainly European beech *Fagus sylvatica*, and non-natural conifer plantations, i.e. Norway spruce *Picea abies* and Scots pine *Pinus sylvestris*. The Steigerwald region (STE) is located in a hilly area (400–520 m a.s.l.) in central Germany (N 49°80′–49°94′; E 10°45′–10°62′) with a large gradient of broadleaf forest use intensity. It is dominated by *F. sylvatica*. The Bavarian Forest National Park region (BAY) is located in a mountainous area (710–1530 m a.s.l.) (N 48° 91′–49°20′; E 13°19′–13°45′). The dominant tree species are *P. abies* and *F. sylvatica*. Half of the area, at the time of data sampling, was dominated by common production forests, while the other half was covered by strictly protected area with intensive natural disturbances or old-growth stands. Thus, the 463 plots included a

long gradient of forest management intensity on the stand scale ranging from unmanaged old-growth forests to intensively managed forests.

**Radar data**. C-band synthetic aperture radar (C-SAR) data acquired by the Sentinel-1 mission throughout Germany, including in our sampling areas, were used in this study. The C-SAR data were acquired in the interferometric wide mode in two polarisations, VV (vertically transmitted, vertically received radar pulse) and VH (vertically transmitted, horizontally received radar pulse), during both ascending and descending orbits. The ground-range-detected high-resolution product (GRDH), with a pixel spacing of 10 m, was downloaded from the ESA Scientific Hub. The Sentinel Application Platforms (SNAP) Sentinel-1 Toolbox software[48] was used in the further processing of the GRDH product to generate $\gamma^0$, defined as the backscatter coefficient normalised by the cosine of the incidence angle. Processing followed the typical pre-processing steps, involving precise orbit determination, removal of thermal and border noise, radiometric calibration to $\beta^0$, defined as the radar brightness coefficient, and radiometric terrain flattening to $\gamma^0$ based on the digital elevation model (DEM) of the Shuttle Radar Topography Mission (SRTM v.4.1), with 1-arc-second resolution. Lastly, a range-Doppler terrain correction was performed, generating $\gamma^0$ with a $10 \times 10$ m pixel spacing also based on the SRTM DEM (see the Supplementary Note 3 for further details and the batch processing graph at https://github.com/So-YeonBae/Sentinel1-Biodiversity and Supplementary Software).

Multiple pixel-wise summary statistics were then calculated over these pre-processed conditions (Supplementary Table 1). The limitation of the short wavelength of Sentinel-1 was overcome by the application of multi-temporal approaches. First, the median values of VV and VH backscatter in a year, during summer and during winter were determined since backscatter varies with the seasonal changes in canopy structure. The difference between the median winter and summer values was then computed to detect the seasonal difference in backscatter. The difference and the ratio between the yearly median values of VV and VH were computed as well, since they are related to the seasonal canopy development cycle[49]. The mean and standard deviation were extracted to characterise the spatial heterogeneity, within a $9 \times 9$ pixel neighbourhood. Further textural variables were then derived by means of the grey-level co-occurrence matrix (GLCM), which specifically considers the spatial arrangement of different neighbourhood pixels[50]. In a GLCM analysis, the contrast group measures the contrast between adjacent pixels, and the orderliness group the orderliness of the neighbourhood pixel values. Both the dissimilarity index in the contrast group and the entropy index in the orderliness group were calculated using window sizes of $9 \times 9$ pixels[51]. All metrics-calculations were performed in R, version 3.4.0[52], using the package glcm[53] with a common discretisation of 32 grey levels (see the Supplementary Note 3 for details and r script at https://github.com/So-YeonBae/Sentinel1-Biodiversity and Supplementary Software).

**ALS data**. ALS data were acquired during leaf-on periods between 2007 and 2018, depending on the region (see Supplementary Table 2). The same pre-processing and metrics-calculation methods were applied over the ALS datasets of all five forest regions. LAStools[54] was used in pre-processing, coordinate transformation, outlier filtering, the classification of the returns into ground and non-ground and the normalisation of the height of the vegetation returns to the height above ground level.

Based on the normalised height, metrics characterising the three-dimensional forest structure were calculated (Supplementary Table 3). The mean and maximum height of the vegetation returns were determined as information on forest maturity, and the standard deviation and coefficient of variation to characterise the vertical variability of the vegetation distribution. Canopy cover as well as the average and variability of the vegetation height are among the most significant predictors of animal species diversity[10]. To characterise the development of canopy sub-layers, the penetration ratios were calculated by dividing the number of returns blocked (or captured) by each sub-layer by the number of returns that reached the layer. Penetration ratios were calculated for three sub-layers: canopy (above 5 m), understory (2–5 m) and regeneration layers (below 2 m). Foliage height diversity was then derived using the Shannon entropy index and the penetration ratios of the three sub-layers. The canopy cover was also determined based on the penetration ratio of the canopy and the understory layer (above 2 m).

Spatial heterogeneity composed of forest gaps were considered by calculating the square-root-transformed total edge length of gap and gap area. Both were determined based on a gap mask raster obtained by mapping pixels with a penetration ratio of the canopy-understory layer <20% and aggregating neighbouring gap pixels into gap features. Gap features smaller than 50 m² or narrower than a perimeter-area-ratio <1.5 were excluded. Lastly, a canopy surface height model (CHM) with a spatial resolution of 1 m was developed according to Khosravipour et al.[55] and using the lidR package[56]. Based on the CHM, the ratio of the canopy surface area to the flat area and the standard deviation of the CHM were calculated.

**Species data sampling**. Bats were recorded using ultrasound detectors or bat-call recorders and analysed with the corresponding software to the species level (see Supplementary Note 1). Repeated point counts were conducted for bird surveys during the breeding seasons. Arthropods were sampled using pitfall traps, flight interception traps and light traps. Vascular plants, bryophytes, lichens and fungi were recorded in spatial subsets of the 1-ha plots. However, the species sampling methods slightly differed between Biodiversity-Exploratories, the Steigerwald Project and the BioKlim-Project in terms of sampling periods, duration and grain size. Hence, the species data were refined to achieve comparability among projects (Supplementary Note 1). We complied with all relevant ethical regulations for animal research. All the records of species, except for arthropods, were conducted by sightings or sound-recording in the field. The methods used in this study to assess arthropod diversity were legally mandated by the field work permits listed in the acknowledgement section and Supplementary Table 11.

**Calculation of biodiversity variables**. Among the various aspects of biodiversity, alpha diversity measures the diversity of species within each plot and beta diversity the difference of species composition among these plots. Gamma diversity is a measure of the overall diversity within a region, a product of the alpha diversity for all the plots within a region and the beta diversity among them, thus often called as regional diversity[57]. At our 1-ha local scale, species richness and phylogenetic diversity, as alpha diversity, and species composition, the base of calculating beta diversity, were investigated. Functional groups were separated based on taxonomy, and the assemblage composition per functional group by NMDS was derived using presence–absence data. The goodness of NMDS for 1–5 dimensions was tested based on the stress value using the function dimcheckMDS in the R package goeveg[58]. The smallest dimension with a stress value < 0.2 was chosen, as done in Clarke[59]. The chosen dimensionality was 2 for plants, lichens, moths, carabid beetles and bats and 3 for all others. NMDS was performed using the function metaMDS in the R package vegan;[60] the Bray–Curtis dissimilarity index and 30 maximum numbers of random starts were used to identify a stable solution.

The number of observed species in each plot was counted and the value log-transformed to calculate species richness. The standardised effect size of the mean phylogenetic distance was determined as phylogenetic diversity which has become influential for an understanding of ecosystem functioning in association with assemblage communities and has been used as a proxy for functional diversity[61,62]. Accordingly, data on published and newly created phylogenies were merged with our community data using the ses.mpd function in picante[63] (see Supplementary Note 2). The null-model approach was applied to correct for species richness, by comparing the diversity value of the observed assemblage per plot with the values of 999 random artificial assemblages containing the same number of species[64]. The species in the 999 assemblages were randomly selected from within each species pool corresponding to that of our five forest regions.

**Canonical correlation analysis**. A CCoA was applied to two datasets, the metrics derived from radar and from ALS. The CCoA represents the observations along new canonical axes that maximise the correlation between two datasets[65]. It was employed in this study to explore the correlation of the radar metrics with the ALS metrics, as the ecological relevance of the latter with respect to 3-D structure and resident animal species diversity has already been investigated. The function cancor in the R package candisc[66] was used in the analysis and all variables in the analysis were standardised. The F-approximations of Wilks' lambda statistic and its significance, the canonical correlations between axes pairs and the RDA-adjusted $R^2$ values were checked to determine whether the canonical axes extracted a considerable variation.

**Modelling biodiversity variables**. Boosted GAMs were applied to all response variables (NMDS axes, log-transformed species richness, and standardised effect size of phylogenetic diversity) with Gaussian error distributions using the function gamboost[67] in the package mboost[67]. Boosted GAMs are suitable for ecological modelling characterised by non-linearity and high collinearity among predictors, which are very common in metrics derived from remote sensing[68]. The predictors chosen in this study had high collinearity; for example, the yearly, summer and winter backscatter of VH showed strong mutual correlations in the radar metrics and the same was determined for the mean and maximum vegetation height in the ALS metrics. Boosted GAMs were chosen because they are suitable for disentangling the effects of variables with collinearity[68] and assumed to be non-linear[40]. The boosted GAMs were implemented with component-wise gradient boosting techniques to optimise parameter estimations and variable selection.

Five-fold cross-validations were performed using the kfold function in the R package dismo[69]; for each one, regions served as the sub-group, achieved by separating the training and test datasets by region and then combining the respective datasets to obtain total training and total test datasets for the five forest regions. To make use of the full range of environmental spaces and species pools covering all the gradients of the five regions, training and test data were extracted from all five regions. For each cross-validation, two models were constructed, one using the radar metrics and the other the ALS metrics. The 40 metrics derived from the radar data and the 13 metrics from the ALS data were used as predictors. Additionally, to account for possible spatial-autocorrelations of plots in the same region and the slightly different sampling years and methods, mixed effects models including the region as a random factor were fitted to determine the comparability with the fixed effects models. However, since the aim of this study was to explore the potential of remote-sensing data in predicting biodiversity over a large area

including our external validation area, and not to test a specific ecological hypothesis, we focused on constructing fixed effects models and adjunctively included mixed effects models.

In each model in each cross-validation, to find the appropriate number of boosting iterations (*mstop*) as a hyper-parameterisation, the *mstop* was increased from 10 to 500 and the corresponding cross-validated estimates of the empirical risk were then checked using the function *cvrisk*; 25 bootstraps were applied in the sampling cross-validation using the function *cv*. Each model was constructed from a training dataset and a hyper-parameter of *mstop* using the function *gamboost* and then applied to a test dataset to examine its predictive performance, using the function *prediction*. The coefficient of determination ($R^2$) between observed values of a test dataset and the predicted values were calculated together with the RMSE. Finally, both the $R^2$ and the RMSE of the five-fold cross-validations were averaged to compare the performances of the different models. For the mixed effects models, in the calculation of $R^2$ and RMSE, only the fixed factors were used to predict the response variables, thereby excluding the variance explained by region (a random factor). Previous studies have shown that the $R^2$ of biodiversity measures increase with sampling size in arthropod samples collected by flight interception and pitfall traps up to a sample size of sufficient individuals[11]. Chao and Jost[70] introduced the sample completeness to standardise the comparability of diversity among communities. Therefore to check the robustness of our results for arthropods against the sampling completeness we re-ran the richness and community composition analyses using a subset of plots with sample coverage more than 90%, 80% and 70%.

As external validations, the first axes of the assemblage composition for birds ($n = 72$) and saproxylic beetles ($n = 91$) outside the five study regions were predicted using radar metrics and fixed effects models, and $R^2$ and RMSE again calculated. In the last step, the assemblage compositions for birds and saproxylic beetles were mapped based on the first NMDS scores across the Bavarian Forest National Park (Supplementary Fig. 20).

**Reporting summary**. Further information on research design is available in the Nature Research Reporting Summary linked to this article.

## Data availability

The data that support the findings of this study are publicly available on the BExIS platform, dataset ID 25206 (https://www.bexis.uni-jena.de/PublicData/PublicDataSet.aspx?DatasetId=25206). The source data underlying Figs. 1 and 4 and Supplementary Figs. 2, 5, 18 and 19 are provided as a Source Data file.

## Code availability

The batch processing configuration file for the SNAP toolbox software and the R script for pixel- and summary statistics are publicly available at https://github.com/So-YeonBae/Sentinel1-Biodiversity and Supplementary Software.

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

## Acknowledgements

This project was financed by the Deutsche Forschungsgemeinschaft (DFG), grant no. MU3621/2–1, KR 3292/2-1 and LE3316/2-1. We thank Hermann Hacker for the determination of moths, Markus Blaschke for lichen sampling in the Steigerwald, and Maria-Barbara Winter for bird sampling in Berchtesgaden. We also thank the managers of the three Exploratories, K. Wells, S. Renner, S. Gockel, K. Wiesner, K. Lorenzen, J. Vogt, A. Hemp and M. Gorke, for their work in maintaining the plots and project infrastructure; C. Fischer and S. Pfeiffer for their support through the central office; M. Owonibi and J. Nieschulze for managing the central database and E. Linsenmair, D. Hessenmöller, D. Prati, I. Schöning, F. Buscot, E.-D. Schulze and the late E. Kalko for their roles in setting up the Biodiversity Exploratories project. This work was funded by the DFG Priority Programme 1374 'Infrastructure-Biodiversity-Exploratories'. Fieldwork permits were issued by the responsible state environmental offices of Baden-Württemberg, Thüringen, Brandenburg and Bayern (see Supplementary Table 11 for details of permits).

## Author contributions

S.B., J.M., S.R.L. conceived the idea of the study and wrote the first manuscript draft. S.B., L.H., P.M., B.F.L., S.W., A.S. analysed the data. S.B., J.M., S.R.L, L.H., P.M, T.N., P.K., M.M.G. contributed to developing the study. L.H., P.S., C.H., C.B., I.D., E.-D.S., F.-S.K., H.C., K.J., M.H., M.F., S.S., S.T., T.G., T.H., W.W.W., J.M. collected data. All authors contributed to the revisions of the manuscript.

## Competing interests
